# Distributed Fixed-Time Secondary Control for MTDC Systems Using Event-Triggered Communication Scheme

**Xiaoyue Zhang** [1], **Xinghua Liu** [1,*] and **Peng Wang** [2]

1    School of Electrical Engineering, Xi'an University of Technology, Xi'an 710048, China;
     zhangxy@stu.xaut.edu.cn
2    School of Electrical and Electronic Engineering, Nanyang Technological University,
     Singapore 639798, Singapore; epwang@ntu.edu.sg
*    Correspondence: liuxh@xaut.edu.cn

**Abstract:** Multi-terminal DC transmission (MTDC) systems have attracted much attention due to their significant advantages in long-distance and high-capacity transmission. To improve their reliability and operation performance, a distributed fixed-time secondary control of frequency restoration and active power sharing is proposed under event-triggered communication, which only depends on the states of each AC grid and its neighbors. By utilizing Lyapunov theory, we prove that the MTDC system with the fixed-time secondary control can be stable in a settling time, and the conditions of the settling time are established for fixed-time algorithms. In addition, we simulate a five-terminal MTDC system in Matlab/Simulink. Several cases of MTDC systems are exhibited to showcase how well the suggested controller works when dealing with load changes and attacks. The comparison of the number of event-triggered instants shows that the proposed control method can effectively reduce communication resources.

**Keywords:** MTDC system; event-triggered communication scheme; distributed fixed-time control





## 1. Introduction

With the exploration and utilization of new energy resources in recent years, the number of distributed energy-generation connections is increasing quickly. Since the settings of distribution network, island power supply, urban power supply and other application fields are expanding, HVDC transmission systems have drawn considerable attention, and the research on them is growing vigorously [1–4]. The frequency reflects the balance relationship between the electromagnetic power and load power of the generator in the AC system [5]. When the active power balance of the AC system is broken caused, say, by a sudden increase or decrease in generator, load, system failure, etc., the frequency of the AC system will fluctuate, in some cases exceeding the maximum-allowable range of frequency, and this threatens the safe and stable operation of the system [6]. Unlike the general AC system, the shielding effect of the DC transmission line makes the MTDC system unable to share the frequency-rotation reserve between each AC grid as well as to realize the optimal allocation of frequency-modulation resources of the whole network [7]. Therefore, designing an effective frequency-recovery control strategy is of great significance to the safe and stable operation of the MTDC system.

In an MTDC system, a hierarchical control scheme is proposed to ensure frequency and voltage recovery as well as active and reactive power control. At present, based on the functions, the MTDC system control architecture can be divided into three layers [8]: the first layer is local control, which regulates the voltage and frequency generated by the inverter [9]. The secondary control is to correct the deviations of voltage and frequency after primary control to eliminate side effects [10]. The third layer is the economic dispatching layer, which makes unified decisions on the overall information of the system [11]. In traditional secondary control for MTDC systems, two conventional methods

are used: global information-based centralized control [12] and local information-based decentralized control [13]. While centralized control relies on communication among all units and the central controller, its performance and reliability are heavily influenced by the communication network's status. In contrast, decentralized control, which operates based on local information and does not require a communication network, is more commonly used and widespread compared to the centralized approach [14]. The distributed control method only exchanges information with neighboring subsystems. Therefore, the stability and anti-interference of the system can be improved by designing the distributed controller [15]. In this paper, we apply the distributed control strategies to MTDC systems; by establishing the communication-network topology between local controllers, the information can be exchanged between local controllers. Previous researches on MTDC systems have shown positive outcomes with hierarchical control. However, there are still areas for improvement that need to be addressed, such as: (1) adjusting the system state based on the predetermined time, (2) reducing information exchange to save communication resources, (3) enhancing the recovery ability in cases of external interference or network attack.

To improve the stability and anti-interference ability of the system, a distributed control strategy is required. We exhibit the relevant research as follows. Li et al. put forward a novel frequency support control method for MTDC systems which considers the safe and efficient active power deficiency or surplus absorption [16]. A distributed cooperative control strategy was proposed in [17], which combines a frequency regulation controller and a load ratio controller to ensure balanced power sharing among stations. Although the previous control strategies effectively achieve their objectives, they overlook the importance of convergence time, which could limit their practical applications. To solve this issue, a new fixed-time control strategy is developed. A set of distributed fixed-time control strategies is utilized to address both the consistency and formation control challenges in multi-agent systems [18–22]. A power system is usually connected by multiple grids, hence the multi-agent system and the power system have a lot in common in control strategies. Wang et al. [23] presented a distributed fixed-time secondary controller that has the capacity to eliminate DC voltage deviation and ensure proportional current sharing. In [24], the control effects of fixed time and finite time in microgrids are compared, and an event-trigger mechanism is adopted in the distributed secondary control. In [25], the secondary voltage control problem for microgrids is transformed into the master–slave consistency problem of single integral multi-agent systems. It can be seen that fixed-time control has been used in microgrids, but few studies have applied it to MTDC systems. In this paper, we apply fixed-time control to MTDC systems, which improves the control performance, anti-interference and robustness of the system.

Many researchers have extensively studied event-triggered communication as a popular approach to minimize communication-transmission energy [26–28]. The event-triggered control strategy can not only reduce data exchange but also eliminate the Zeno phenomenon. Zhang et al. [29] introduced the event-triggered distributed hybrid control scheme aimed at ensuring the safe and economically efficient operation of an integrated energy system. In [30], feeder-impedance mismatch and precise power sharing in isolated microgrids are solved via a distributed event-triggered power sharing control strategy. Wang et al. [31] proposed a distributed event-triggered secondary control scheme in microgrids, which can can reduce the communication frequency according to the change in microgrid state. Compared with MG systems, the communication burden of the MTDC system is greater, hence adopting the event-trigger mechanism for MTDC system control can effectively reduce the communication cost. Table 1 displays a comparative analysis of our work and the primary existing studies are listed.

**Table 1.** Comparative Summary of this and previous publications.

| Refs. | System Types | Control Scheme | Transmission Form |
|-------|-------------|----------------|-------------------|
| [18] | Multi-agent system | Distributed fixed-time control | Event-triggered mechanism |
| [30] | Microgrids | Distributed secondary control | Event-triggered mechanism |
| [23] | Microgrids | Distributed fixed-time secondary control | Time-triggered mechanism |
| [25] | Microgrids | Distributed fixed-time secondary control | Event-triggered mechanism |
| [12] | MTDC transmission system | Centralized secondary control | Time-triggered mechanism |
| [13] | MTDC transmission system | Decentralized secondary control | Time-triggered mechanism |
| [17] | MTDC transmission system | Distributed secondary control | Time-triggered mechanism |
| This study | MTDC transmission system | Distributed fixed-time secondary control | Event-triggered mechanism |

From previous research, we know that the fixed-time control strategy has faster dynamics and the convergence time is independent of the initial value of the system. Event-trigger mechanisms can reduce the communication costs. Consequently, this paper introduces a distributed secondary fixed-time control approach for MTDC systems using an event-triggered strategy. The proposed method successfully achieves frequency recovery and active power sharing in a predetermined time. This paper has the following main contributions:

(1) The paper introduces a distributed secondary fixed-time control scheme utilizing an event-triggered mechanism to achieve frequency restoration and active power sharing. This approach circumvents the communication loop problem in [32] and prevents Zeno behavior in [33]. Different from [34], the corresponding upper limit of convergence time is determined.

(2) Through the strict stability principle, the sufficient conditions for the stability of event-trigger parameters and control-gain constraints in MTDC systems are proved. Compared with previous results in [10,34], the suggested approach demonstrates strong potential in effectively utilizing limited communication resources and control energies

(3) Compared with the existing methods in [22,23,25], the fixed-time algorithms presented in this paper exhibit a robust resistance to unknown external bounded interference, thereby enhancing the transient and overall robust performance of the MTDC system. Additionally, a set of simulation results validates the effectiveness of the proposed algorithm.

*Notations*

$\text{sig}(x)^y = \text{sign}(x)|x|^y$ and $\text{sign}(\cdot)$ represents the sign function. $\lambda_1^*$ and $\lambda_N^*$ denote the minimum and maximum eigenvalues of the Laplacian matrix, respectively.

The remainder of this paper is organized as follows. Section 2 presents some needed lemmas and preliminaries. Section 3 presents the formulation of the problem. The distributed fixed-time secondary controllers are designed and exhibited in Section 4. Numerical studies are presented in Section 5. The paper ends with concluding remarks in Section 6.

## 2. Preliminaries

### 2.1. Graph Theory

In this paper, we consider the MTDC system's communication topology between AC grids represented by an undirected graph. The details of the graph are defined as follows. The graph vertices represent the AC grids and communication links are constructed as graph edges. Let $\mathcal{G}(\mathcal{V}, \mathcal{E}, \mathcal{A})$ denote an undirected graph, where $\mathcal{G}^\omega(\mathcal{V}^\omega, \mathcal{E}^\omega, \mathcal{A}^\omega)$ and $\mathcal{G}^P(\mathcal{V}^P, \mathcal{E}^P, \mathcal{A}^P)$ denote frequency and an active power communication network, respectively. $\mathcal{V} \subset \{v_i | i = 1, \ldots, N\}$ is the set of nodes and $\mathcal{E} \subset \mathcal{V} \times \mathcal{V}$ is the set of edges. The adjacency matrix is denoted by $\mathcal{A} = [a_{ij}] \in R^{N \times N}$. $i$ and $j$ are denoted as $(i, j)$ when

they are directly connected and supposing there is a communication channel between them. If edge $(i, j) \in \mathcal{E}$, then $a_{ij} = a_{ji} > 0$; otherwise, $a_{ij} = a_{ji} = 0$; therefore $a_{ij}$ can be referred to as communication parameters. Next, $\mathcal{N}_i = \{j \in \mathcal{V} | (i, j) \in \mathcal{E}\}$ denotes the set of neighbors of node $i$. The Laplacian matrix of graph $\mathcal{G}$ is defined as $\mathcal{L} = [l_{ij}] \in R^{N \times N}$. Notably, with $\mathcal{D} = \text{diag}\{\mathcal{D}_1, \mathcal{D}_1, \ldots, \mathcal{D}_N\}$ where $\mathcal{D}_i = \sum_{j \in \mathcal{N}_i} a_{ij}$, we denote $\mathcal{L} = \mathcal{D} - \mathcal{A}$.

In addition to the above nodes, we define a virtual leader AC bus as $v_0$ and denote the diagonal matrix as $\mathcal{Q} = \text{diag}\{q_1, q_2, \ldots, q_N\}$, where $q_i$ is the connection weight between the AC bus $i$ and the leader node. We define the undirected graph with the virtual leader node as $\bar{\mathcal{G}}$.

### 2.2. Lemma and Definition

**Definition 1.** *Suppose that the origin is the equilibrium point of the following system:*

$$\begin{cases} \dot{x}(t) = f(t, x(t)) \\ x(0) = x_0 \end{cases} \tag{1}$$

*If it is asymptotically stable, the origin of (1) is globally finite-time stable and the stability time $T > 0$. If $\exists T_{\max} > 0$, then $T \leq T_{\max}$ for any initial conditions; it is fixed-time stable.*

**Lemma 1.** *Assume $\dot{x} = f(t, x), x(0) = x_0$ with $f(t, 0) = 0$. Suppose there is an unknown nonlinear function $V : R^n \to R_+ \cup \{0\}$ that (1) $V(x) = 0 \Leftrightarrow x = 0$; (2) any solution $x(t)$ satisfies the following inequality*

$$D^+ V(x(t)) \leq -(\alpha V^p(x(t)) + \beta V^q(x(t)))^k,$$

*where $k > 0, pk < 1, qk > 1$. As a result, it is determined that the system achieves global stability and the maximum convergence time is limited to:*

$$T(x_0) \leq 1 / \left[\alpha^k(1 - pk)\right] + 1 / \left[\beta^k(qk - 1)\right], \forall x_0 \in R^n.$$

**Lemma 2.** *Let $\varsigma_1, \varsigma_2, \ldots, \varsigma_n \geq 0, 0 < \varphi \leq 1, \phi > 1$. Then*

$$\sum_{i=1}^{n} \varsigma_i^{\varphi} \geq \left(\sum_{i=1}^{n} \varsigma_i\right)^{\varphi}, \sum_{i=1}^{n} \varsigma_i^{\varphi} \geq n^{1-\phi} \left(\sum_{i=1}^{n} \varsigma_i\right)^{\varphi}.$$

**Lemma 3.** *Since G is a connected undirected graph, the Laplacian matrix L of G has the following properties: $x^T L x = \frac{1}{2} \sum_{i=1}^{n} \sum_{j=1}^{n} a_{ij} (x_i - x_j)^2$, for any $x = [x_1, x_2, \ldots, x_n]^T$, the matrix L is symmetric positive semidefinite, which means L has 0 and 1 as eigenvalues. Then the matrix L has an eigenvalue $\lambda_2 (\lambda_2 > 0)$, and it is the second smallest; furthermore, if $1^T x = 0$, then $x^T L x \geq \lambda_2 x^T x$.*

## 3. Problem Formulation

This section briefly depicts the topology of MTDC systems. Then, the hierarchical control structures including droop control and secondary control levels are introduced and the main objectives of this paper are put forward.

### 3.1. MTDC Systems

This paper considers an MTDC system composed of multiple converters, with indexes of $i = 1, \ldots, n$. Each converter is connected to an AC grid. The DC sides of all converters are connected to the MTDC system. The topology is given in Figure 1.

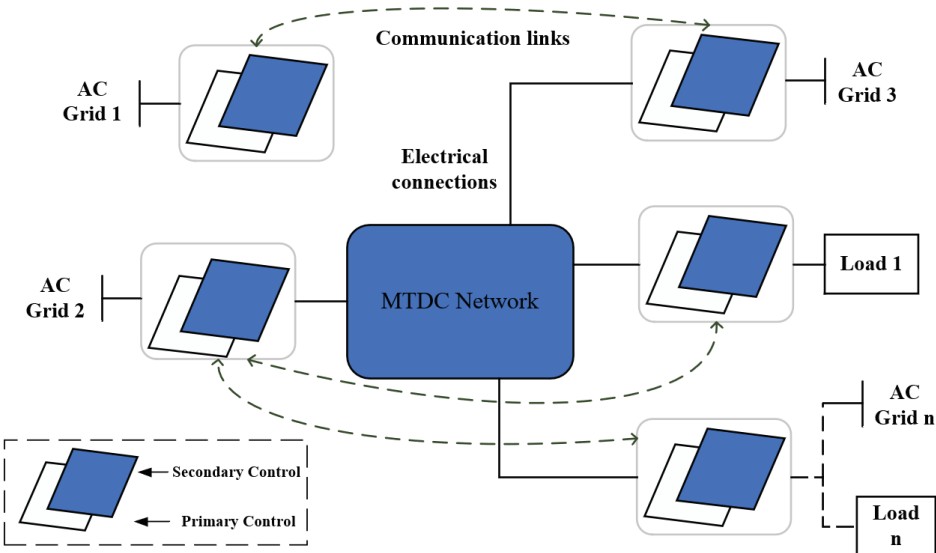

**Figure 1.** Control architecture of the MTDC system.

### 3.2. Control Hierarchy of MTDC System

(1) *Primary Droop Control Layer*: Ref. [35] provides an insightful discussion on the modeling and control of MTDC converters. In particular, frequency control changes the rated power in the converter's active power loop. The primary droop control adjusts the output voltage signal based on power to achieve self-regulation and automatic power distribution by modifying frequency and voltage amplitude. Figure 2 illustrates the commonly used droop control law:

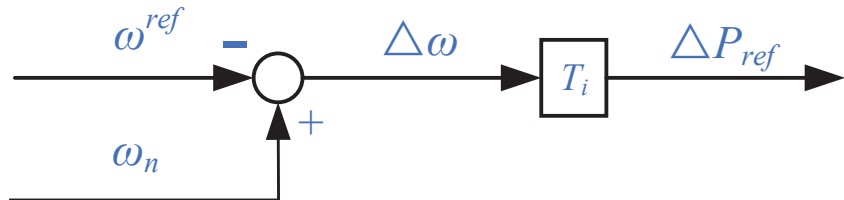

**Figure 2.** Traditional primary droop control.

$$P_i - P_d = -k_{\omega_i}\left(\omega_i - \omega^{ref}\right). \tag{2}$$

Then, we have

$$\omega_i = \omega^{ref} - T_i^p P \tag{3}$$

where $T_i^p = -\frac{1}{k_{\omega_i}}$, $P = P_i - P_d = \triangle P_{ref}$.

(2) *Secondary Control Layer*: Using feedback linearization technology and taking the input disturbance into account, the continuous-time dynamics of each AC bus can be expressed as follows

$$\dot{\omega}_i(t) = u_i^{\omega}(t) + \alpha_i^{\omega}(t), \tag{4}$$

$$\dot{P}_i(t) = u_i^P(t) + \beta_i^P(t), \tag{5}$$

where max $\left\| \alpha_i^{\omega}(t) \right\| \leq \lambda_{\alpha}^{\omega}$ and max $\left\| \beta_i^{P}(t) \right\| \leq \kappa_{\beta}^{P}$ with $\lambda_{\alpha}^{\omega} \geq 0$ and $\kappa_{\alpha}^{p} \geq 0$. Then, the set values of frequency can be calculated via (2) as

$$
\begin{aligned}
\omega_i^n(t) &= \int \left( \dot{\omega}_i + T_i^P \dot{P}_i \right) ds \\
&= \int (u_i^{\omega} + \alpha_i^{\omega}) ds + T_i^P \left( u_i^P + \beta_i^P \right) ds.
\end{aligned}
\tag{6}
$$

*3.3. Control Objectives*

It is clear from (2) that at the steady state, since $P_i(t) \neq 0$, the AC grid's frequency will deviate from the rated frequency, Hence, the distributed secondary fixed-time controllers $u_i^{\omega}(t)$ and $u_i^P(t)$ under an event-triggered scheme are proposed to achieve the following objectives:

(1)　*Frequency Regulation*: to ensure the frequency recovery,

$$
\lim_{t \to T_{\omega}} \left( \omega_i(t) - \omega^{ref} \right) = 0
\tag{7}
$$

(2)　*Active Power Sharing*: to distribute active power flexibly in fixed-time while applying the event-trigger scheme,

$$
\lim_{t \to T_P} \left( T_i P_i(t) - T_j P_j(t) \right) = 0, \forall i \neq j
\tag{8}
$$

Unlike the finite-time stability algorithm, fixed-time stability control can adjust the parameters of controllers to realize the required convergence rate despite the initial value. Therefore, the fixed-time control strategy with an event-triggered algorithm is proposed in this paper and will be introduced in the next section.

## 4. Fixed-Time Control with Event-Triggered Communication Scheme

In actual operation, for the sake of effectively controlling MTDC systems, a shorter convergence time is expected. We propose a distributed fixed-time secondary control scheme which enables governors to design the frequency recovery time off-line in advance subjects with respect to the requirements.

*4.1. Fixed-Time Controller Design with Event-Triggered Scheme*
4.1.1. Frequency Recovery

The control input of agent $u_i^{\omega}$ under the event-triggered strategy with continuous communications can be designed as

$$
u_i^{\omega}(t) = k_{1\omega}(\tilde{\omega}_i(t_s))^{\gamma} + k_{2\omega}(\tilde{\omega}_i(t_s))^{\eta} + k_{3\omega}\text{sign}(\tilde{\omega}_i(t_s))
\tag{9}
$$

and

$$
\tilde{\omega}_i(t_s) = \sum_{j \in \mathcal{N}_i} a_{ij}\left( \omega_j(t_{s'}) - \omega_i(t_s) \right) + g_i\left( \omega^{ref} - \omega_i(t_s) \right),
\tag{10}
$$

where $0 < \gamma < 1, \eta > 1, k_{i\omega} > 0$. As the $i$th AC grid has reached the rated frequency $\omega^{ref}$, $g_i = 1$ or $g_i = 0$.

The event-triggered measurement error can be denoted as

$$
\begin{aligned}
\chi_i^{\omega}(t) &= k_{1\omega}(\tilde{\omega}(t_s))^{\gamma} - k_{1\omega}(\tilde{\omega}(t))^{\gamma} + k_{2\omega}(\tilde{\omega}(t_s))^{\eta} \\
&\quad - k_{2\omega}(\tilde{\omega}(t))^{\eta} + k_{3\omega}\text{sign}(\tilde{\omega}(t_s)) - k_{3\omega}\text{sign}(\tilde{\omega}(t)).
\end{aligned}
\tag{11}
$$

Then, with (10) and (11), frequency dynamics (3) can be expressed as

$$
\begin{aligned}
\dot{\omega}_i(t) &= k_{1\omega}(\tilde{\omega}_i(t))^{\gamma} + k_{2\omega}(\tilde{\omega}_i(t))^{\eta} \\
&\quad + k_{3\omega}\text{sign}(\tilde{\omega}_i(t)) + \chi_i^{\omega}(t) + \alpha_i^{\omega}(t).
\end{aligned}
\tag{12}
$$

We define $\delta_i^\omega(t) = \omega_i^t - \omega^{ref}$; the frequency-restoration error is

$$
\begin{aligned}
\dot{\delta}_i^\omega(t) = & k_{1\omega}\big(\tilde{\delta}_i^\omega(t)\big)^\gamma + k_{2\omega}\big(\tilde{\delta}_i^\omega(t)\big)^\eta \\
& + k_{3\omega}\big(\tilde{\delta}_i^\omega(t)\big) + \chi_i^\omega(t) + \alpha_i^\omega(t),
\end{aligned}
\tag{13}
$$

where

$$
\tilde{\delta}_i^\omega(t) = \sum_{j \in \mathcal{N}_i^\omega} a_{ij}\big(\delta_j^\omega(t) - \delta_i^\omega(t)\big) - g_i \delta_i^\omega(t).
\tag{14}
$$

The time sequence is denoted as

$$
t_{k+1}^i = \inf\left\{ t > t_k^i \right\} | \mathcal{Z}_i^\omega(c_i^\omega, \chi_i^\omega, \tilde{\omega}_i(t)) \geq 0 |,
\tag{15}
$$

where

$$
\mathcal{Z}_i^\omega(c_i^\omega, \chi_i^\omega, \tilde{\omega}_i(t)) = |\chi_i^\omega(t)| - c_i^\omega |\tilde{\omega}_i(t)|^\gamma.
\tag{16}
$$

#### 4.1.2. Active Power Sharing

Similar to the above frequency-recovery algorithm, the active power allocation algorithm is designed as

$$
u_i^P(t) = k_{1P}\big(\tilde{P}_i(t_s)\big)^\gamma + k_{2P}\big(\tilde{P}_i(t_s)\big)^\eta + k_{3P}\text{sign}(\tilde{P}_i(t_s))
\tag{17}
$$

and

$$
\tilde{P}_i(t_s) = \sum_{j \in \mathcal{N}_i} a_{ij}\big(P_j(t_{s'}) - P_i(t_s)\big),
\tag{18}
$$

where $k_{3P} \geq \kappa_\beta^P$ and $k_{iP} > 0$. Similar to the above frequency control, the active power estimate error is set as $\chi_i^P(t) = P_i(t_s) - P_i(t)$. The AC bus is triggered when the following condition is satisfied

$$
t_{k+1}^i = \inf\left\{ t > t_k^i \,\Big|\, \mathcal{Z}_i^P\Big(c_i^P, \chi_i^P, \tilde{P}(t)\Big) \geq 0 \right\},
\tag{19}
$$

where

$$
\mathcal{Z}_i^P\Big(c_i^P, \chi_i^P, \tilde{P}(t)\Big) = \left|\chi_i^P(t)\right| - c_i^P |\tilde{P}_i(t)|^\gamma.
\tag{20}
$$

The overall control framework is presented in Figure 3 for better understanding.

**Remark 1.** *From Equations (9) and (17), we can see that the right side of these equations can be divided into two parts. The first two terms guarantee the frequency recovery and active power distribution in fixed time, the third term ensures that the controller is not disturbed by external disturbances.*

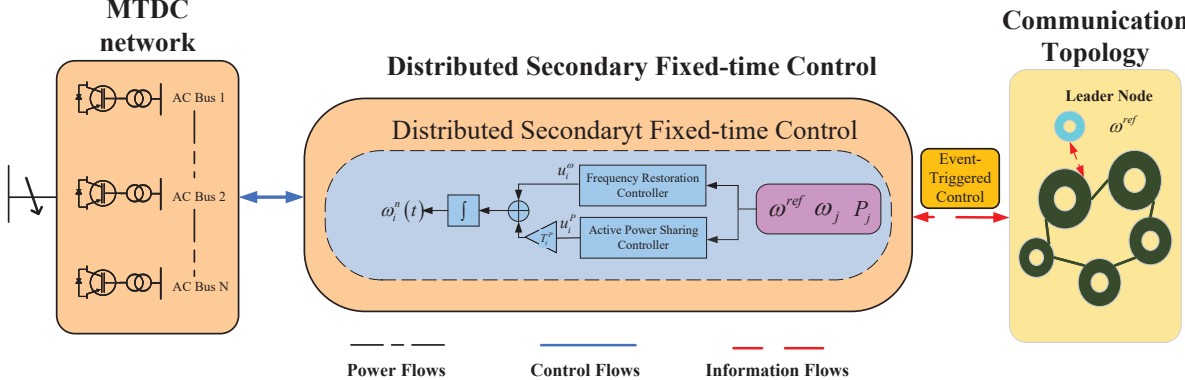

**Figure 3.** The overall control framework.

*4.2. Stability Analysis*

**Theorem 1.** *Consider MTDC system (2) and control input (9) under communication rule (16), the distributed fixed-time controller can realize frequency recovery regardless of initial value if $k_{1\omega} > c_i^\omega > 0$; the settling time is determined by*

$$
T_{\max}^\omega \leq \frac{2}{k_{2\omega}\left(2\lambda_1^\omega\right)^{\frac{1+\eta}{2}}(1-\eta)}
$$
$$
+ \frac{2}{(k_{1\omega} - c_{\max}^\omega)N^{\frac{1-\gamma}{2}}\left(2\lambda_1^\omega\right)^{\frac{1+\gamma}{2}}(\gamma - 1)}
\tag{21}
$$

The objective of achieving predetermined convergence time for frequency recovery is accomplished using the following Lyapunov function.

$$
V_\omega(\delta_\omega) = \frac{1}{2}\delta_\omega^T(\mathcal{L}_\omega + G_n)\delta_\omega,
\tag{22}
$$

where $\delta_\omega = \left[\delta_1^\omega, \delta_2^\omega, \ldots, \delta_N^\omega\right]^T$ and $G_n = \text{diag}\{g_1, g_2, \ldots g_n\}$.

The Lyapunov potential function $V_\omega(\delta_\omega)$ is considered reasonable because matrix $\mathcal{L}_\omega + G_n$ is positive semidefinite. The derivate of $V_\omega(\delta_\omega)$ is given by

$$
\dot{V}_\omega(\delta_\omega) = \delta_\omega^T(\mathcal{L}_\omega + G_n)\dot{\delta}_\omega
$$
$$
= \sum_{i=1}^N \left(\sum_{j=1}^N a_{ij}^\omega\left(\delta_i^\omega - \delta_j^\omega\right) + g_i^\omega \delta_i^\omega\right)\dot{\delta}_i^\omega
$$
$$
= -\sum_{i=1}^N \tilde{\delta}_i^\omega \dot{\delta}_i^\omega.
\tag{23}
$$

Substituting (14) into (24), one derives

$$
\dot{V}_\omega(\delta_\omega) = -\sum_{i=1}^N \tilde{\delta}_i^\omega \dot{\delta}_i^\omega
$$
$$
= -k_{1\omega}\sum_{i=1}^N \tilde{\delta}_i^\omega \left(\tilde{\delta}_i^\omega\right)^\gamma - k_{2\omega}\sum_{i=1}^N \tilde{\delta}_i^\omega \left(\tilde{\delta}_i^\omega\right)^\eta
$$
$$
- k_{3\omega}\sum_{i=1}^N \tilde{\delta}_i^\omega \text{sign}\left(\tilde{\delta}_i^\omega\right) - \sum_{i=1}^N \tilde{\delta}_i^\omega \chi_i^\omega - \sum_{i=1}^N \tilde{\delta}_i^\omega \alpha_i^\omega.
\tag{24}
$$

Due to $\left|\tilde{\delta}_i^\omega(t)\right| = \tilde{\delta}_i^\omega(t)\text{sign}\left(\tilde{\delta}_i^\omega(t)\right)$, we have

$$
\dot{V}_\omega(\delta_\omega) \leq -k_{1\omega}\sum_{i=1}^N \left(\tilde{\delta}_i^\omega\right)^{1+\gamma} - k_{2\omega}\sum_{i=1}^N \left(\tilde{\delta}_i^\omega\right)^{1+\eta}
$$
$$
- k_{3\omega}\left|\tilde{\delta}_i^\omega\right| + \sum_{i=1}^N \left|\tilde{\delta}_i^\omega\right|\left|\chi_i^\omega\right| + \sum_{i=1}^N \left|\tilde{\delta}_i^\omega\right|\left|\alpha_i^\omega\right|.
\tag{25}
$$

With (16) and (10), we can obtain

$$
\dot{V}_\omega(\delta_\omega) \leq -k_{1\omega}\sum_{i=1}^N \left(\tilde{\delta}_i^\omega\right)^{1+\gamma} - k_{2\omega}\sum_{i=1}^N \left(\tilde{\delta}_i^\omega\right)^{1+\eta}
$$
$$
- k_{3\omega}\left|\tilde{\delta}_i^\omega\right| + c_{\max}^\omega\sum_{i=1}^N \left|\tilde{\delta}_i^\omega\right|\left|\tilde{\omega}_i\right|^\gamma + \sum_{i=1}^N \left|\tilde{\delta}_i^\omega\right|\left|\alpha_i^\omega\right|.
\tag{26}
$$

When $\left|\tilde{\delta}_i^\omega\right| = |\tilde{\omega}_i(t)|$, we have

$$
\begin{aligned}
\dot{V}_\omega(\delta_\omega) = &-k_{1\omega} \sum_{i=1}^{N} \left(\tilde{\delta}_i^\omega\right)^{1+\gamma} - k_{2\omega} \sum_{i=1}^{N} \left(\tilde{\delta}_i^\omega\right)^{1+\eta} \\
&- k_{3\omega} \left|\tilde{\delta}_i^\omega\right| + c_{\max}^\omega \sum_{i=1}^{N} \left|\tilde{\delta}_i^\omega\right| \left|\tilde{\delta}_i^\omega\right|^\gamma + \sum_{i=1}^{N} \left|\tilde{\delta}_i^\omega\right| |\alpha_i^\omega|.
\end{aligned}
\tag{27}
$$

Through the previous calculation, the above inequality can be written as

$$
\begin{aligned}
\dot{V}_\omega(\delta_\omega) \leq &-k_{1\omega} \sum_{i=1}^{N} \left(\left(\tilde{\delta}_i^\omega\right)^2\right)^{\frac{1+\gamma}{2}} - k_{2\omega} \sum_{i=1}^{N} \left(\left(\tilde{\delta}_i^\omega\right)^2\right)^{\frac{1+\eta}{2}} \\
&- k_{3\omega} \sum_{i=1}^{N} \left|\tilde{\delta}_i^\omega\right| + c_{\max}^\omega \sum_{i=1}^{N} \left(\left(\tilde{\delta}_i^\omega\right)^2\right)^{\frac{1+\gamma}{2}} \\
&+ \sum_{i=1}^{N} \left|\tilde{\delta}_i^\omega\right| |\alpha_i^\omega| \\
= &-(k_{1\omega} - c_{\max}^\omega) \sum_{i=1}^{N} \left(\left(\tilde{\delta}_i^\omega\right)^2\right)^{\frac{1+\gamma}{2}} \\
&- k_{2\omega} \sum_{i=1}^{N} \left(\left(\tilde{\delta}_i^\omega\right)^2\right)^{\frac{1+\eta}{2}} + \sum_{i=1}^{N} \left|\tilde{\delta}_i^\omega\right| |\alpha_i^\omega| \\
&- k_{3\omega} \sum_{i=1}^{N} \left|\tilde{\delta}_i^\omega\right|.
\end{aligned}
\tag{28}
$$

According to Lemma 1, we have

$$
\begin{aligned}
\dot{V}_\omega(\delta_\omega) \leq &-(k_{1\omega} - c_{\max}^\omega) N^{\frac{1-\gamma}{2}} \sum_{i=1}^{N} \left(\left(\tilde{\delta}_i^\omega\right)^2\right)^{\frac{1+\gamma}{2}} \\
&- k_{2\omega} \sum_{i=1}^{N} \left(\left(\tilde{\delta}_i^\omega\right)^2\right)^{\frac{1+\eta}{2}} - (k_{3\omega} - \lambda_\alpha^\omega) \sum_{i=1}^{N} \left|\tilde{\delta}_i^\omega\right|.
\end{aligned}
\tag{29}
$$

With the help of $\lambda_\alpha^\omega > k_{3\omega}$, (29) becomes

$$
\begin{aligned}
\dot{V}_\omega(\delta_\omega) \leq &-(k_{1\omega} - c_{\max}^\omega) N^{\frac{1-\gamma}{2}} \sum_{i=1}^{N} \left(\left(\tilde{\delta}_i^\omega\right)^2\right)^{\frac{1+\gamma}{2}} \\
&- k_{2\omega} \sum_{i=1}^{N} \left(\left(\tilde{\delta}_i^\omega\right)^2\right)^{\frac{1+\eta}{2}}.
\end{aligned}
\tag{30}
$$

Based on the above analysis and the properties of the undirected graph, it is concluded that

$$
\begin{aligned}
\dot{V}_\omega(\delta_\omega) \leq &- (k_{1\omega} - c_{\max}^\omega) N^{\frac{1-\gamma}{2}} 2(\lambda_1^\omega)^{\frac{1+\gamma}{2}} (V_\omega(\delta_\omega))^{\frac{1+\gamma}{2}} \\
&- k_{2\omega} (2\lambda_1^\omega)^{\frac{1+\eta}{2}} (V_\omega(\delta_\omega))^{\frac{1+\eta}{2}}.
\end{aligned}
\tag{31}
$$

Utilizing fixed-time stability in [36] as the foundation, the proposed control ensures that the frequency-tracking error converges to zero and the settling time can be determined using Equation (22). The proof is thereby concluded.

**Theorem 2.** *Consider MTDC system (2) and control input (17) under communication rule (20), the distributed fixed-time controller can realize active power sharing regardless of initial value if $k_{2P} > c_i^P > 0$; the settling time is determined by*

$$T_{\max}^P \leq \frac{2}{k_{2P}\left(2\lambda_1^P\right)^{\frac{1+\eta}{2}}(1-\eta)} + \frac{2}{(k_{1P} - c_{\max}^P)N^{\frac{1-\gamma}{2}}\left(2\lambda_1^P\right)^{\frac{1+\gamma}{2}}(\gamma - 1)} \tag{32}$$

Consider system (4), to realize the predeterminate convergence time of active power sharing, the following Lyapunov function is employed

$$V_P(\delta_P) = \frac{1}{2}\delta_P^T(\mathcal{L}_P + G_n)\delta_P. \tag{33}$$

The derivation is similar to the proof of Theorem 1. For simplicity, it is omitted here.

**Remark 2.** *According to the above formula, different parameters in the controller algorithm will affect the convergence speed. We take the distributed-frequency fixed-time recovery algorithm (10) as an example; the parameter $k_{i\omega}, \gamma, \eta, c_i^P, \lambda_1^\omega$ will affect the convergence time of frequency recovery. The specific impact will be verified in the next chapter of simulated results.*

## 5. Illustrative Examples

In this section, the control strategy's validity is confirmed through numerical simulations conducted in Matlab/Simulink. The tested MTDC system is depicted in Figure 4, with specific parameters provided in Table 2. The AC grid parameters are obtained from [37]. The simulation focuses on examining the frequency restoration capability and power-sharing aspects of the proposed control strategy.

### 5.1. Load Change Circumstances

In this chapter, we mainly verify the effectiveness of the presented controller and its performance under the load-change circumstances. We divided the whole process into four steps:

(1)  Within 0∼5 s, the primary control is activated and all loads work normally.
(2)  Within 5∼15 s, both the primary control and secondary control are activated, and all loads function normally.
(3)  Within 15∼20 s, only the primary control is activated, with Loads 2 and 3 operating normally, while Loads 1 and 4 are disconnected.
(4)  Within 20∼30 s, both the primary control and secondary control are operational, and Loads 2 and 3 function normally, while Loads 1 and 4 have been disconnected.

The simulated results are shown in Figures 5 and 6. Within 0∼5 s and 15∼20 s, only primary control is activated, and with the frequency restoration and active power distribution, the control objectives we proposed are not realized. In 5∼15 s and 20∼30 s, after the secondary control is applied, the system demonstrates the ability to restore the frequency to 50 Hz and distribute active power proportionally, regardless of whether load changes occur. This validates the effectiveness and robustness of the proposed controller. As is evident from Figures 5 and 6, the frequency-restoration process occurs swiftly and with accurate estimation within 8 s, significantly less than $T_\omega = 15.36$ s. Additionally, the active power allocation is achieved in approximately 10.4 s, which is considerably faster than $T_P = 27.75$ s.

In conclusion, the simulated outcomes indicate that the proposed controller can achieve fixed-time terminal frequency synchronization of the AC grid and effectively distribute active power in MTDC systems.



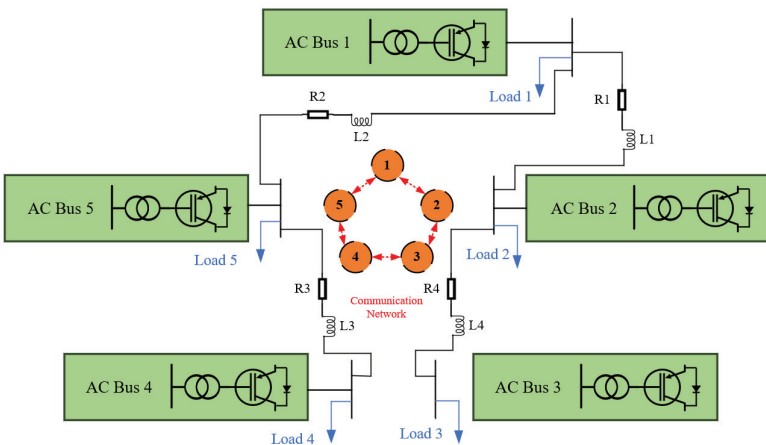

**Figure 4.** Topology of the MTDC system.

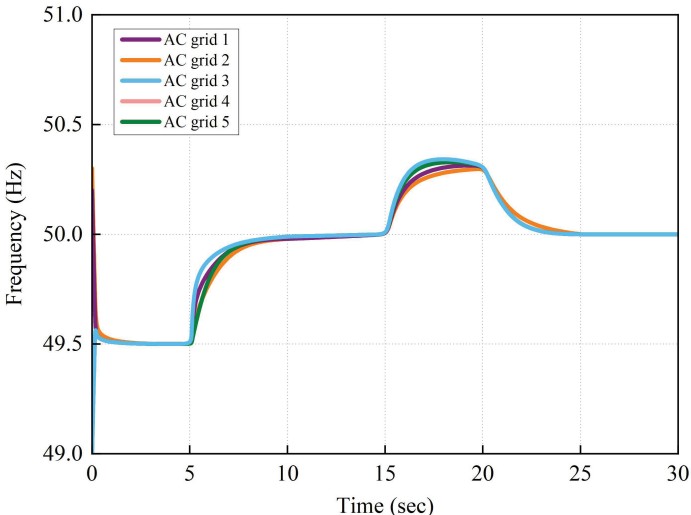

**Figure 5.** Frequency restoration with proposed controller under load change.

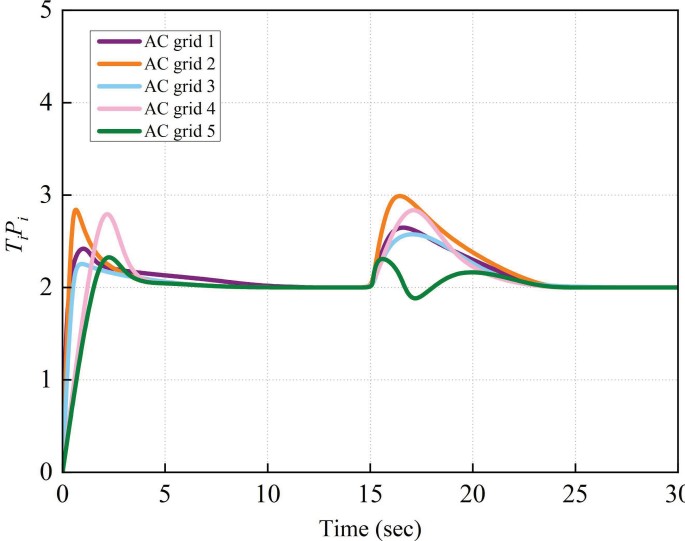

**Figure 6.** Active power sharing with proposed controller under load change.

**Table 2.** Parameters of test systems.

| Parameter | Symbol | Value |
|---|---|---|
| AC System Frequency | $f$ | 50 Hz |
| AC Nominal Voltage | $V_s$ | 10 kV (L to L) |
| AC Side Resistance | $R$ | $1 \times 10^{-4}$ |
| AC Side Inductance | $L$ | 0.705 mH |
| DC Nominal Voltage | $V_d$ | 20 kV |
| DC Side Capacitance | $C$ | $10^{-5}$ F |
| DC Line Resistance | $R_{line}$ | $1.45 \times 10^{-2}$ |
| DC Line Inductance | $L_{line}$ | 0.165 mH/km |

### *5.2. Performance under Attacks*

The MTDC system may be attacked during operation. The attacker can interfere with the communication channel, destroy the equipment, prevent the equipment from sending messages, attack the routing protocol, etc. To examine the effectiveness of the presented controller under attack, we conducted the following experiments: during 0~30 s, the primary and secondary control are both activated; the communication lines between AC grid 3 and AC grid 4 are disconnected at $t = 5$ s and reconnected at $t = 5$ s (to simulate an attack on line); AC grid 5 is unplugged at $t = 10$ s and then plugged back in at $t = 20$ s (to simulate an attack on the node). The simulation results are shown in Figures 7 and 8.

It can be seen that the curve of frequency and active power fluctuates when AC grid 5 unplugged and plugged back in. For all that, the presented distributed controller can realize the frequency recovery and achieve active power sharing. The above results confirm the anti-attack ability of the presented controller. This is of great significance for the safe operation of MTDC systems.

### *5.3. Some Comparison Results*

To best exhibit the performance of the proposed controller, we conduct the following comparative simulations under the same scenario in the first case during 0~15 s.

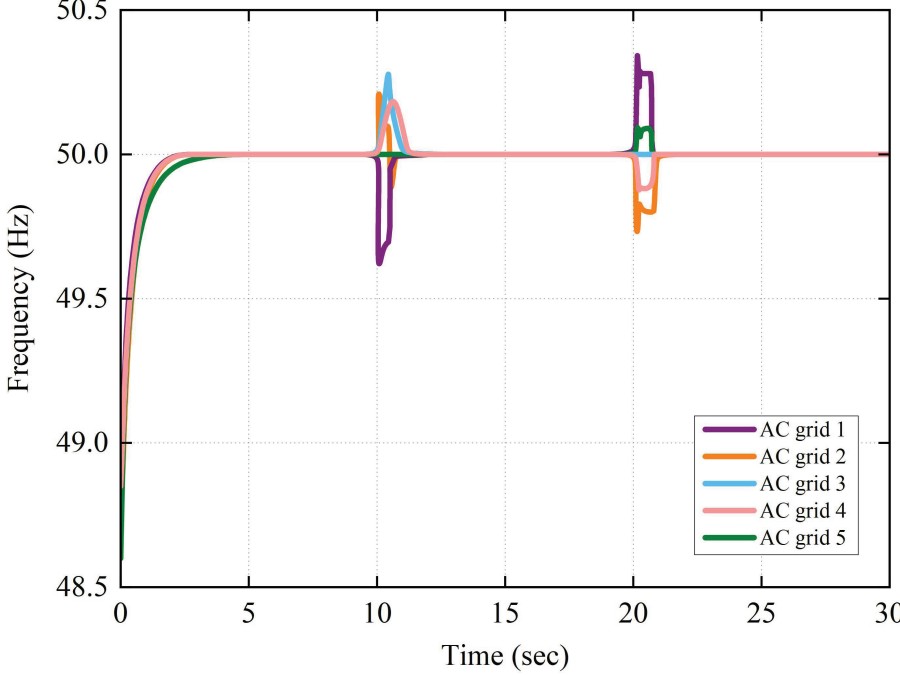

**Figure 7.** Frequency restoration with proposed controller under attack.

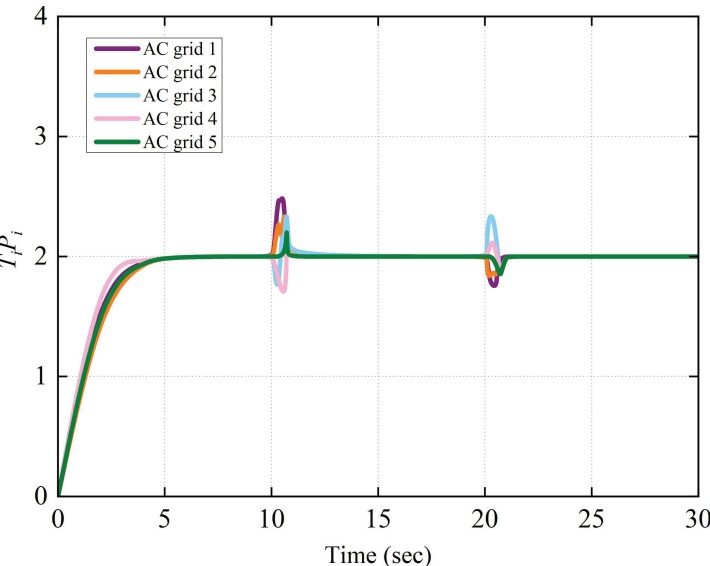

**Figure 8.** Active-power-sharing curve with proposed controller under attack.

### 5.3.1. Convergence Performance with Different Parameters

As we can seen from Theorems 1 and 2, $c_i$ constrains the convergence time and affects its value. Therefore, we take $c_i = 0.05$, $c_i = 0.1$, $c_i = 6$ in a comparative experiment; the results are shown in Figure 9.

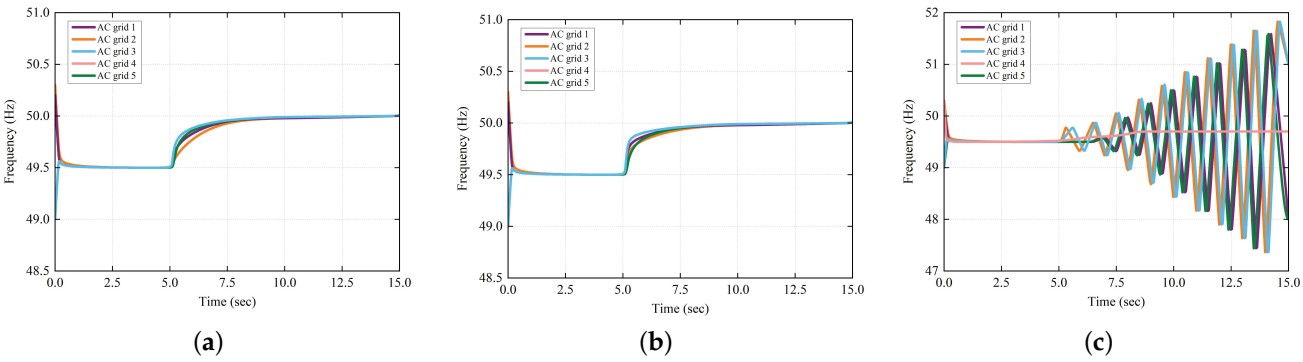

| (a) | (b) | (c) |

**Figure 9.** Frequency restoration with different $c_i$. (**a**) Frequency restoration with $c_i = 0.1$. (**b**) Frequency restoration with $c_i = 0.05$. (**c**) Frequency restoration with $c_i = 6$.

The experimental results show that when $c_i = 0.05$, the convergence speed is faster than when $c_i = 0.1$. When $c_i$ is too large $c_i = 6$, the frequency results diverge and the proposed controller cannot achieve the objective. Therefore, we need to carefully select the value of $c_i$ to make the controller achieve the optimal performance.

### 5.3.2. Quantity of Communication

The quantities of communication instants for the proposed distributed controller with and without the event-triggered strategy for AC grids are shown in Figure 10. The number of communications between AC grids with and without the event-triggered strategy are represented in blue, orange and yellow. Through the column, it is obvious that with the event-triggered strategy, the above strategy can effectively reduce the communication of frequency recovery and active power distribution.

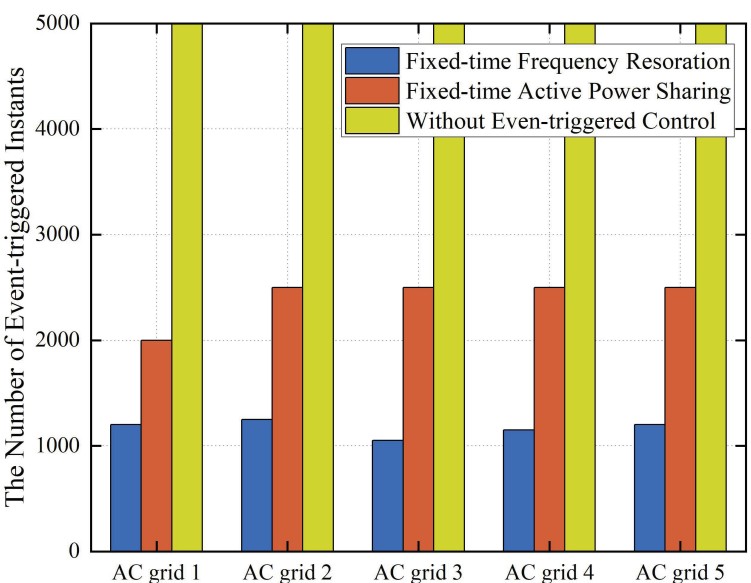

**Figure 10.** The number of event-triggered instants.

### 5.3.3. Comparative Study with Previous Literature

In this section, we compare the proposed fixed-time frequency control with the distributed secondary control in the literature [34] and the finite-time control in the literature [38] to verify that our proposed method has better convergence. Before 4 s, only the droop control is in use, and it is evident that the frequency does not return to the rated frequency. At 4 s, the secondary control is activated; the simulation results are displayed in Figure 11. The proposed fixed-time controller restores the frequency to the rated frequency in just 1.55 s, whereas the linear consensus control and finite-time control achieve frequency restoration at 1.9 s and 1.75 s, respectively. Clearly, compared to the approaches presented in previous literature, the fixed-time control proposed in this paper operates more efficiently.

In this section, several scenarios of MTDC systems are examined to showcase the effectiveness and validity of the proposed controller in the presence of load changes and attacks. The comparison of event-triggered instances reveals that the proposed control method significantly reduces the utilization of communication resources while maintaining its effectiveness. Moreover, on comparison with the existing literature, the simulated results show that the proposed controller has better performance.

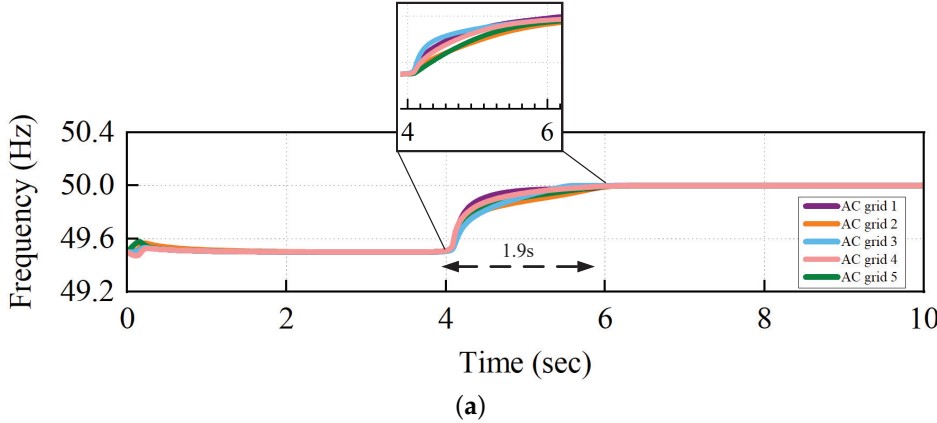

(**a**)

**Figure 11.** *Cont.*

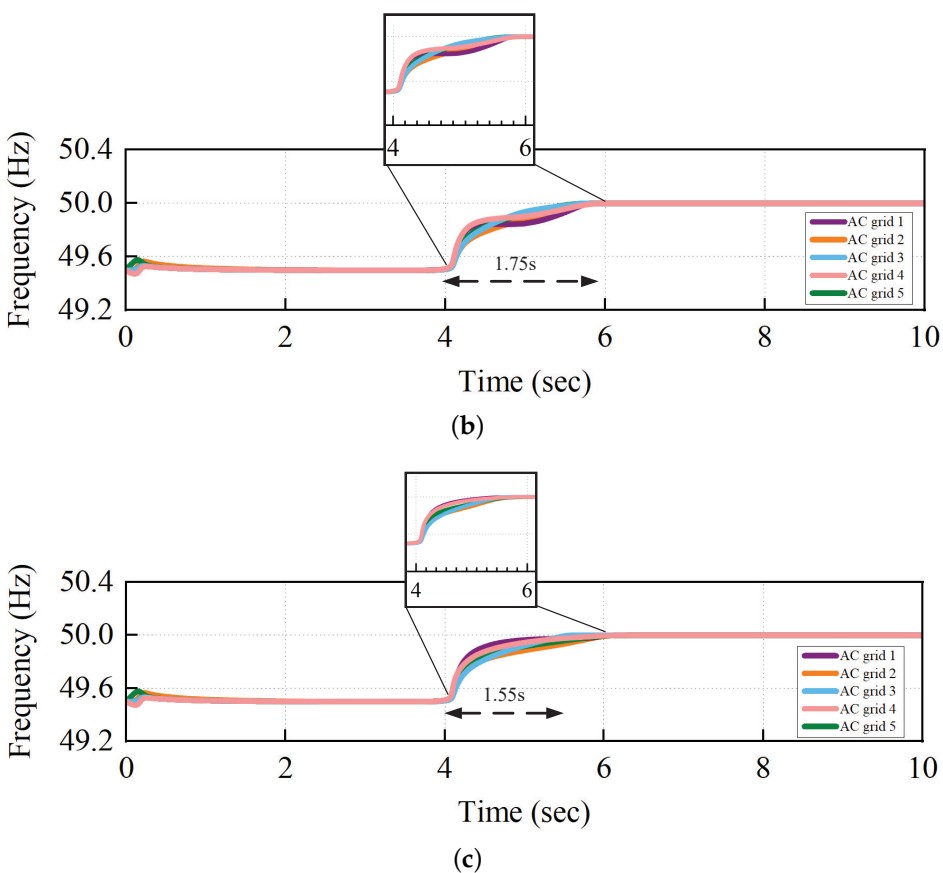

**Figure 11.** Comparison of frequency recovery via different methods. (**a**) Consensus algorithm. (**b**) Finite-time algorithm. (**c**) Fixed-time algorithm.

## 6. Conclusions

The main focus of this paper is to explore distributed resilient control methods for active power allocation and frequency restoration in MTDC systems. The proposed controller takes the fixed-time control into account, which can complete the objectives in a certain time without considering the initial value. Moreover, in order to save communication resources between AC grids, the event-triggered strategy was considered. Based on the above, the proposed controller realized frequency recovery and active power distribution in a fixed time and reduced communication burdens. To verify the proposed theorems, an MTDC system was built in Matlab/Simulink R2022b software to carry out a series of experiments to verify the performance of the proposed controller, including the following circumstances: load change, under attack and parameter change. In future work, we will move a step further to design a distributed prescribed-time controller considering time-delay for MTDC systems, which cannot only resist faults and communication constraints but also achieves regulations under uncertain nonlinearities.

**Author Contributions:** Conceptualization and writing—original draft preparation, X.Z.; writing—review and editing, X.L.; supervision, P.W. All authors have read and agreed to the published version of the manuscript.

**Funding:** This research was funded in part by the National Natural Science Foundation of China under Grant U2003110 and in part by the High Level Talents Plan of Shaanxi Province for Young Professionals.

**Data Availability Statement:** Not applicable.

**Conflicts of Interest:** The authors declare no conflict of interest.

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
