# Peer review of "Distributed Fixed-Time Secondary Control for MTDC Systems Using Event-Triggered Communication Scheme"

_processes, doi:10.3390/pr11082329_

Round 1

Reviewer 1 Report

The manuscript contains novel contributions, is well-written and well-organised. Please find below some parts that need to be improved to make it ready for publication:

1) In this reviewer's opinion, to better motivate the need of fixed-time control of MTDC systems, the beginning of the Introduction should be extended to provide more examples of emerging application scenarios where such methodologies would be of fundamental value. For instance, next-generation environmental monitoring systems relying on different infrastructures would significantly benefit from efficient and distributed energy generation connections. In this respect, some pointers to recent and interesting literature on this field could be added, such as "Toward integrated large-scale environmental monitoring using WSN/UAV/Crowdsensing: A review of applications, signal processing, and future perspectives", Sensors, 2022. But also other additional use cases could be added as well.

2) Some additional discussions in the stability analysis conducted in Section 4.2 would be helpful.

3) In Table I, there are other existing approaches for distributed, possibly fixed-time control, but in Section 5 no comparisons with any of them is provided. Why? Please provide a clear justification.

4) The beginning of Section 5.3 seems incomplete. Could the authors please double-check it?

N/A

Author Response

Please see the attachment and thank you veyr much.

Reviewer 2 Report

The text discusses a proposed distributed fixed-time secondary control system for multi-terminal DC transmission (MTDC) systems. The control system aims to improve reliability and operation performance by restoring frequency and sharing active power. The proposed system relies on event-triggered communication based on the states of each AC grid and its neighbors. The use of Lyapunov theory proves the stability of the MTDC system with the fixed-time secondary control. The settling time conditions for fixed-time algorithms are established. A simulation of a 5-terminal MTDC system in Matlab/Simulink demonstrates the effectiveness and validity of the proposed controller under load changes and attacks. The proposed control method reduces the number of event-triggered instants, effectively reducing communication resources.
the intention of the paper is to propose a new distributed fixed-time secondary control for Multi-terminal DC (MTDC) systems using event-triggered communication. The aim is to improve the reliability and operation performance of the MTDC systems for long-distance and high-capacity transmission. The proposed controller is designed to restore the frequency and allocate active power in the system using Lyapunov theory, and its effectiveness is demonstrated through MATLAB/Simulink simulations under different scenarios, including load changes and attacks. 

The results are as follows:

- The proposed distributed fixed-time secondary control algorithm can restore the frequency and share the active power effectively in MTDC systems under different scenarios, including load changes and attacks.

- The conditions of settling time are established for fixed-time algorithms in MTDC systems.

- The stability of the proposed controller is proven through Lyapunov theory.

- The proposed controller is tested through MATLAB/Simulink simulations, and the results show that the controller is effective in improving the reliability and operation performance of MTDC systems while maintaining their stability.

- The number of event-triggered actions required by the proposed controller is reduced in comparison to some existing methods, which can reduce the communication overhead and improve the efficiency of the system.

Author Response

Please see the attachemnt and thank you very much.

Reviewer 3 Report

This paper proposes the application of distributed fixed-time secondary control for frequency restoration and active power sharing in multi-terminal DC transmission systems, in order to improve their reliability and operational performance. The stability of the proposed control is verified using Lyapunov theory.

The topic is interesting and the paper is well-organized. However, the proposed control strategy does not appear to introduce a significant improvement compared to conventional control methods. The performance of the finite-time algorithm is better than the proposed algorithm in restoring the frequency to the rated value in less time at all terminals. As shown in Figure 11, it took about 1.75 seconds for the finite-time algorithm to restore the frequency of all terminals, while the proposed algorithm needed about 2 seconds to achieve the same result.

Author Response

Please see the attachment and thank you very much.

Round 2

Reviewer 1 Report

The authors correctly addressed all my comments.

N/A

Reviewer 3 Report

The paper can be accepted in its present form.